# The Effectiveness of Taurolidine Antimicrobial Locks in Preventing Catheter-Related Bloodstream Infections (CRBSIs) in Children Receiving Parenteral Nutrition: A Case Series

**DOI:** 10.3390/antibiotics13090847

**Published:** 2024-09-05

**Authors:** Galina Ling, Shalom Ben-Shimol, Siham Elamour, Raouf Nassar, Eyal Kristal, Rotem Shalev, Gadi Howard, Baruch Yerushalmi, Slava Kogan, Moshe Shmueli

**Affiliations:** 1The Faculty of Health Sciences, Ben-Gurion University of the Negev, Beer-Sheva 84101, Israel; galinali@clalit.org.il (G.L.); sihame@clalit.org.il (S.E.); raoufna@clalit.org.il (R.N.); eyalkr2@clalit.org.il (E.K.); rotemsha@clalit.org.il (R.S.); gadiho@clalit.org.il (G.H.); baruchye2@clalit.org.il (B.Y.); slavako@clalit.org.il (S.K.); mosheshm@post.bgu.ac.il (M.S.); 2The Pediatric Day-Care Unit, Soroka University Medical Center, Beer-Sheva 84101, Israel; 3The Pediatric Infectious Disease Unit, Soroka University Medical Center, Beer-Sheva 84101, Israel

**Keywords:** pediatrics, taurolidine antimicrobial lock, parenteral nutrition, catheter-related bloodstream infections (CRBSIs)

## Abstract

Introduction: We assessed the efficacy of taurolidine lock (TL) in preventing catheter-related bloodstream infections (CRBSIs) and related hospitalizations in children with parenteral nutrition (PN) in the home setting. Methods: This study is a retrospective case series study. All children with intestinal failure in a single center in southern Israel who were administered PN and treated with TL between 2017 and 2024 were included. The rates of CRBSI episodes, related hospitalizations and pathogen distribution in the pre-TL and post-TL periods were compared. Results: Overall, 14 patients were included. The median pre-TL and post-TL periods were 990 and 1260 days, respectively. The rate of CRBSI episodes due to bacterial infection per 1000 days declined by 45%, from 6.2 to 3.7, with *p* = 0.0008, while fungal CRBSI rates were low (<10% of all positive cultures) and did not decline significantly. Similarly, the hospitalization episode rate per 1000 days declined by 41%, from 7.6 to 4.5, with *p* = 0.001. Conclusions: Taurolidine lock treatment for children with central-line PN resulted in a substantial decrease in CRBSI episodes and related hospitalizations.

## 1. Introduction

Central-line access is essential for parenteral nutrition (PN) in patients with intestinal failure [1]. Catheter-related bloodstream infections (CRBSIs) are common complications of total parenteral nutrition (TPN) [2]. The rate of CRBSIs varies significantly in different populations, impacted by patient demographics, socioeconomic status, disease, catheter type, and many other factors [3,4]. These infections frequently necessitate hospital admissions for intravenous antibiotics, a delay in primary disease treatment, early catheter removal, or even intensive care unit admissions due to severe sepsis [5]. Consequently, improved catheter care and the resulting reduction in CRBSIs should lower mortality, decrease treatment costs, and shorten hospital stays [6]. These efforts are especially crucial for patients with impaired bowel function who depend on long-term catheters for parenteral feeding, mostly at home [1]. Avoiding contamination that would lead to subsequent catheter colonization is essential in decreasing the risk of CRBSIs [7].

For the past two decades, various antibiotic and antiseptic solutions have been used to flush or lock catheter lumens [8]. The catheter lock technique involves filling the lumen with an antimicrobial solution to dwell while not in use [9]. Antibiotics like vancomycin, gentamicin, ciprofloxacin, and others, either alone or combined, and antiseptic solutions, such as ethanol, trisodium citrate, and taurolidine, have been employed [5]. 

Taurolidine, a derivative of taurine used in medicine since the 1970s, is safe and nontoxic at therapeutic doses, with no bacterial resistance reported [2,10,11,12]. Rapidly metabolized into taurine, carbon dioxide, and water, taurolidine works by binding to bacterial cell walls, causing damage and preventing biofilm formation inside catheters [13,14]. 

Present guidelines recommend TL and other antibiotic locks for patients with recurrent catheter infections, reducing the risk of infection by about fourfold. Patients treated with taurolidine showed significantly lower reinfection rates compared to those treated with heparin [5,15,16]. 

The current literature lacks comprehensive guidelines for preventing CRBSIs in children, particularly in those with impaired bowel function. At the Soroka University Medical Center (SUMC) pediatric day care unit, we evaluate and monitor these patients, adhering to aseptic techniques, regularly checking for infections, and educating healthcare providers and caregivers on proper catheter care [17,18]. Recently, we have incorporated the use of TL into our prevention strategy, as studies have shown it significantly reduces infection rates [5,10,15,19,20].

The most common bacteria associated with CRBSIs include coagulase-negative staphylococci (CoNS), *Staphylococcus aureus*, Enterococci, Candida species, *Escherichia coli*, Klebsiella species, and *Pseudomonas aeruginosa* [21,22]. These organisms are frequently found in the skin flora or introduced during catheter care, underscoring the need for rigorous aseptic practices to reduce infection rates. Although TL has demonstrated efficacy against these pathogens, there remains limited comparative knowledge regarding its impact on the bacterial profile of CRBSI.

In this case series, we aimed to assess the efficacy of TL in preventing catheter-related infections in children with PN in the home setting. First, we analyzed the number of infections and follow-up hospitalizations among our patients before and after starting TL use. Second, we identified the bacterial profiles of infections in each patient before and after TL use to evaluate its effectiveness against different bacteria.

## 2. Methods

### 2.1. Study Design

This retrospective case series study assessed all children with intestinal failure who were administered PN and treated with TL at the SUMC pediatric day care unit between 2017 and 2024 (*n* = 14). Patient characteristics, culture results, and clinical parameters were obtained from the medical files and entered into a computerized database. For each patient, the follow-up period was divided into pre- and post-treatment with TL, with each patient serving as their own control, being included in both periods. Taurolidine lock has been incorporated into our center since June 2020. Therefore, we set the follow-up period from January 2017, three years prior to the incorporation of the treatment (pre-treatment period), to June 2024 (post-treatment period). During these follow-up months, episodes of CRBSI were counted and analyzed, including culture results and hospitalizations. Patient age was recorded at the time of TL administration.

This study was approved by the SUMC review board committee. Due to its retrospective nature, a waiver of informed consent was granted.

### 2.2. Setting and Study Population

The SUMC is the only 1200-bed hospital in the Negev district of southern Israel, providing primary and referral health care services to the entire population of the region (~200,000 children under 18 years). Over 95% of the children in the Negev region are served by the SUMC, as medical insurance for children is universal and free of charge. Practically all children in southern Israel with intestinal failure, many suffering from prematurity complications such as necrotizing enterocolitis (NEC), and others diagnosed with Hirschsprung disease, intestinal atresia, or congenital diarrhea and enteropathy, are treated at the SUMC pediatric day care unit. 

### 2.3. Inclusion Criteria

Patients included in the study were children with intestinal failure, treated with PN via a central line after 2017, and administered TL. All patients used a catheter (tunneled or implantable port) for central venous access and were thus eligible in case of developing CRBSIs. All patients were trained on aseptic techniques by a specialized nurse during admission to our hospital before starting home PN. This training followed a standardized protocol in our center.

### 2.4. Taurolidine Lock Treatment Protocol

Patients received TauroLock™ (TauroPharm GmbH, Waldbüttelbrunn, Germany) [23,24]. We used TauroLock™ (1.35% taurolidine + 4% citrate) due to its availability and to enhance thrombus safety. While guidelines do not require anticoagulants for parenteral nutrition devices, citrate complements the antibacterial effects of taurolidine and aligns with institutional protocols. All patients were guided to follow the manufacturer’s instructions that accompany the particular venous vascular access product utilized. Specific catheter lock volumes are associated with each device. The treatment was provided by the caregiver daily after parenteral nutrition. The treatment steps are briefly described as follows: 1. Flush the device with 10 mL of saline to ensure patency. 2. Withdraw TauroLock™ from the container using an appropriate syringe. 3. Instill TauroLock™ slowly (not more than 1 mL per second; for infants and children less than two years of age, not more than 1 mL per 5 s) into the access device in a quantity sufficient to fill the lumen completely. 4. Keep TauroLock^TM^ in the device on daily basis until the subsequent treatment. 5. Flush the device with 10 mL of saline.

The SUMC pediatric day care unit team followed the manufacturer’s instructions for the specific fill volume or specified fill volume during implantation. Apart from the switch to a TL in the assigned patients, there was no alteration in the protocol for catheter care during the study period.

### 2.5. Diagnosis of CRBSI Episodes

#### 2.5.1. Positive Cultures

A diagnosis of a CRBSI episode was made whenever a patient presented with a positive blood culture. According to the Infectious Diseases Society of America (IDSA) definition, the growth of microorganisms had to be shown from at least two blood culture sets taken from the catheter and peripheral blood, as there is no infection of another site [25]. Although the gold standard for diagnosing CRBSIs is a culture of the catheter tip, this was not included as a diagnostic criterion since we aimed to preserve the line whenever possible [22]. Clinical characteristics such as fever were not considered, as blood cultures might be taken at a low level of suspicion in patients with central lines, and interpretation based on clinical features alone might be inconclusive. We counted the number of months with positive cultures, treating two successive positive cultures within a month as a single episode.

#### 2.5.2. Hospitalization Episode

A hospitalization episode was recorded when a culture was obtained during a hospital stay of at least two days, with or without a CRBSI diagnosis. This was done to detect the overall number of hospitalizations that raised clinical suspicion of catheter-related infection, and not necessarily those determined as CRBSI episodes, and consequently to assess changes in the number of hospitalization episodes pre- and post-TL treatment. Several hospitalizations within a month were counted as a single hospitalization episode. If the hospitalization was longer than 30 days, it was counted as two months of hospitalization episodes.

### 2.6. Blood Culture Result Profile 

Bacterial cultures were systematically analyzed by collecting blood samples from patients presenting with clinical signs of a bloodstream infection. These samples were incubated and monitored for bacterial growth. Each isolated bacterial strain was identified using standard microbiological techniques, including Gram staining, biochemical tests, and when necessary, molecular methods. The frequency and type of bacterial species were recorded and compared across the pre- and post-TL treatment periods. 

We categorized the pathogens into several groups: Enterobacteria, other/environmental Gram-negative bacteria, Gram-positive bacteria (including coagulase-negative Staphylococci [CONS], *Staphylococcus aureus*, and others), and different types of fungi. This categorization was based on the disease characteristics of the patients being followed. We assumed that patients with intestinal failure might have a higher rate of Enterobacteria bacteremia, while others, such as oncology patients, might have predominantly Gram-positive bacteremia due to the prevalence of skin pathogens [24]. These data were then statistically analyzed to determine any significant differences in bacterial profiles associated with the use of taurolidine lock.

### 2.7. Statistical Methods

The data collected were documented using summary tables. Continuous variables with a normal distribution were presented as means and standard deviations. Descriptive population categorical variables were presented as counts and percentages of the total. Categorical variables were tested using Pearson’s χ^2^ test for contingency tables. All statistical tests and confidence intervals, as appropriate, were performed at α = 0.05 (two-sided) and presented with their 95% confidence interval when appropriate, using IBM SPSS version 29.

## 3. Results

During the study period, 14 patients met the inclusion criteria and were analyzed. Twelve (86%) patients were of Bedouin ethnicity, and ten (71%) were males. The mean (±standard deviation) age at administrating TL was 7.3 ± 4.8 years. 

Background illnesses included short bowel syndrome in seven (50%) patients, tufting enteropathy in three (21%) children, neurogenin 3 deficiency in two (14%) children, and jejunal atresia in one (7%) child. Additionally, one (7%) child was diagnosed with hypoparathyroidism, retardation, and dysmorphism (HRD).

### 3.1. CRBSI Episodes

The pre-TL period ranged between 30 days and 1500 days (median of 990 days, mean of 846 ± 611 days). During this period, the range of CRBSI episodes was 0 and 28 episodes, representing the minimum and maximum number of episodes observed (median of 2.5 episodes, mean of 5.3 ± 7.7 days) (Table 1).

The post-TL period ranged between 120 days and 1560 days (median of 1260 days, mean of 1082 ± 497 days). During this period, the range of CRBSI episodes was 0 and 13 episodes (median of 1.5 episodes, mean of 3.6 ± 4.2 days).

Overall, the CRBSI episode rate per 1000 days declined by 45%, from 6.2 to 3.7, with *p* = 0.0008.

### 3.2. Hospitalization Episodes

During the pre-TL period, the range of hospitalization episodes was 0 and 23 episodes (median of 4.5 episodes, mean of 6.4 ± 6.8 days) (Table 2).

During the post-TL period, the range of hospitalization episodes was 0 and 14 episodes (median of 3.5 episodes, mean of 4.9 ± 4.6 days).

Overall, hospitalization episodes rate per 1000 days declined by 41%, from 7.6 to 4.5, *p* = 0.001.

### 3.3. Pathogen Distribution

During the pre-TL period, the rates of Gram-negative and Gram-positive bacteria in CRBSI episodes were similar, at 6.1 and 6 per 1000 days, respectively, while the rate of fungal CRBSI episodes was 0.6 per 1000 days. Among Gram-negative cases, Enterobacter and Klebsiella species were the most common pathogens, while coagulase-negative Staphylococci (CONS) predominated the Gram-positive group (Table 3).

During the post-TL period, the rates (per 1000 days) of Gram-negative bacteria CRBSI episodes declined by 47%, with *p* = 0.0008. Similarly, Gram-positive bacteria CRBSI episodes declined by 43%, with *p* = 0.003. In contrast, the fungal CRBSI rate did not decline significantly.

The overall pathogen (groups) episode rates declined by 44%, from 12.7 to 7.1 episodes per 1000 days, with *p* < 0.001.

## 4. Discussion

In this retrospective case series study, we report our experience administering TL to children with intestinal failure who rely on parenteral nutrition. The initiation of TL at our center resulted in nearly a 50% decrease in the number of CRBSI episodes among these patients, providing strong evidence for the effectiveness of this treatment. This reduction was observed in both Gram-negative bacteria (which constituted 50% of all pathogens detected) and Gram-positive bacteria. However, this effect was not observed in fungal infections, possibly driven by the small number of fungal infections in our cohort. Additionally, the incorporation of TL resulted in a reduced rate of hospitalizations, regardless of a culture-proven CRBSI diagnosis.

The effectiveness of the TL regimen in the prevention of CRBSIs was previously presented in several studies, mostly involving adults [3,21,26,27,28,29]. A meta-analysis published in 2020, which included four randomized controlled trials (RCTs) comparing the effects of TL in preventing CRBSIs in pediatric patients, indicated a statistically significant reduction in the total number of CRBSI episodes with TL compared to the control [30]. We focused on children suffering from intestinal failure, since this population usually experiences higher rates of CRBSIs, resulting in health impairment and high costs [26,31]. Our results support the suitability of this prevention strategy for children with intestinal failure, demonstrating an overall reduction of 45% in the relative risk of developing CRBSIs. Similarly, this reduction was evident when we compared the number of hospitalizations before and after TL treatment, indicating that TL decreased not only positive culture rates but also the number of episodes leading to hospitalization among these patients.

One of the main considerations with TL administration is the implementation of treatment and the mechanical difficulties, as TL is mostly administered by a caregiver in the home setting [32,33]. Children with intestinal failure are even more at risk of infections, as they often have stomas or other implanted devices, providing a solid base for pathogens from the gastrointestinal tract or skin. Our results highlight the fact that even in these cases, TL antimicrobial treatment is effective.

Children with intestinal failure are more exposed to CRBSIs originating from the bacterial population of the gastrointestinal tract [34,35]. These pathogens are mostly Gram-negative, such as E. coli, Klebsiella, and Enterobacter species. Previous studies primarily focused on the efficacy of the TL regimen when treating children with oncological diseases who have central lines, demonstrating a different population of bacteria, mostly Gram-positive from the skin [24]. In our research, we demonstrated that the effectiveness of TL was evident for both Gram-negative and Gram-positive bacteria, including environmental Gram-negative bacteria, such as Pseudomonas and Acinetobacter species. As for fungal infections, which were found at a low rate in our cohort (less than 10% of all positive cultures), the treatment resulted in a moderate decline (rate reduction of 0.66) without statistical significance. It is possible that due to the small number of episodes, we were unable to demonstrate the true effect of TL antimicrobial treatment against fungi.

We acknowledge several limitations. First, this is a single-center, retrospective study. Second, in contrast to other studies, we chose not to compare the TL regimen to other controls, such as heparin alone, but to compare pre- and post-treatment periods for rates of CRBSIs, hospitalization, and pathogen distribution. We believe that this comparison could reduce the influence of possible confounders. The number of patients followed in this study was relatively small, although it is similar to previous research cohorts in the field.

## 5. Conclusions

In conclusion, in this retrospective case series, we reported a single-center experience of a nearly 50% decrease in the number of CRBSI episodes when administering TL to children with intestinal failure. This finding was consistent across different types of bacteria and provided supportive evidence for the utility of TL in patients treated with PN. An extension of the TL regimen should be considered for other patient populations, such as those treated with other types of central venous access devices, including short-term and intermediate-term devices.

## Figures and Tables

**Table 1 antibiotics-13-00847-t001:** Catheter-related bloodstream infection (CRBSI) episodes per central line (CL) days.

Patient	Episodes Pre-TAUR-LOCK (*n*/days)	Episodes Post-TAUR-LOCK (*n*/days)	*p* Value	Episodes per 1000 Days (pre-TL)	Episodes per 1000 Days (Post-TL)	RR (95% CI)
1	0/30	0/120	1	0 per 1000	0 per 1000	-
2	3/120	6/480	0.4	25 per 1000	12.5 per 1000	0.51 (0.13–2.00)
3	5/510	3/420	0.74	9.8 per 1000	7.1 per 1000	0.73 (0.18–3.04)
4	2/570	1/930	0.56	3.5 per 1000	1.1 per 1000	0.31 (0.03–3.38)
5	14/1440	4/1230	0.06	9.7 per 1000	3.3 per 1000	0.34 (0.11–1.02)
6	10/750	2/1560	0.0004	13.3 per 1000	1.3 per 1000	0.10 (0.02–0.44)
7	28/1500	13/1080	0.21	18 per 1000	12 per 1000	0.65 (0.34–1.25)
8	6/1410	1/1290	0.13	4.3 per 1000	0.8 per 1000	0.18 (0.02–1.52)
9	3/1350	9/1470	0.15	2.2 per 1000	6.1 per 1000	2.74 (0.74–10.12)
10	0/1350	0/1530	1	0 per 1000	0 per 1000	-
11	0/30	1/540	1	0 per 1000	1.9 per 1000	-
12	2/1230	1/1530	0.59	1.6 per 1000	0.7 per 1000	0.40 (0.04–4.43)
13	0/60	10/1530	1	0 per 1000	6.5 per 1000	-
14	1/1500	0/1440	1	0.7 per 1000	0 per 1000	-
TOTAL	74/11,850	51/15,150	0.0008	6.2 per 1000	3.7 per 1000	0.55 (0.38–0.77)

**Table 2 antibiotics-13-00847-t002:** Hospitalization episodes per central line (CL) days.

Patient	Episodes Pre-TAUR-LOCK (*n*/days)	Episodes Post-TAUR-LOCK (*n*/days)	*p* Value	Episodes per 1000 Days (pre-TL)	Episodes per 1000 Days (Post-TL)	RR (95% CI)
1	0/30	0/120	1	0 per 1000	0 per 1000	-
2	3/120	8/480	0.47	25 per 1000	17 per 1000	0.67 (0.18–2.50)
3	2/510	3/420	0.66	3.9 per 1000	7.1 per 1000	1.82 (0.30–10.82)
4	2/570	1/930	0.56	3.5 per 1000	1.1 per 1000	0.31 (0.03–3.38)
5	10/1440	4/1230	0.28	6.9 per 1000	3.3 per 1000	0.47 (0.15–1.49)
6	10/750	8/1560	0.04	13.3 per 1000	5.1 per 1000	0.39 (0.15–0.98)
7	23/1500	14/1080	0.74	15.3 per 1000	13 per 1000	0.85 (0.44–1.64)
8	13/1410	3/1290	0.02	9.2 per 1000	2.3 per 1000	0.25 (0.07–0.89)
9	12/1350	9/1470	0.51	8.9 per 1000	6.1 per 1000	0.69 (0.29–1.63)
10	0/1350	0/1530	1	0 per 1000	0 per 1000	-
11	0/30	1/540	1	0 per 1000	1.9 per 1000	-
12	9/1230	4/1530	0.09	7.3 per 1000	2.6 per 1000	0.36 (0.11–1.16)
13	0/60	12/1530	1	0 per 1000	7.8 per 1000	-
14	6/1500	1/1440	0.13	4 per 1000	0.7 per 1000	0.17 (0.02–1.45)
TOTAL	90/11,850	68/15,150	0.001	7.6 per 1000	4.5 per 1000	0.59 (0.43–0.81)

**Table 3 antibiotics-13-00847-t003:** Positive culture episodes per central line (CL) days.

		Episodes Pre-TAUR-LOCK (*n*/days)	Episodes Post-TAUR-LOCK (*n*/days)	*p* Value
				RR (95% CI)
Enterobacteriaceae	Enterobacter	23	6	0.0002
	Klebsiella	21	13	0.05
	E-coli	8	7	0.63
	ESBL	0	0	-
	Other	2	4	0.91
Other Gram-negative bacteria	Pseudomonas	6	8	0.85
	Acinetobacter	5	6	0.84
	Other	7	5	0.47
All Gram-negative bacteria		72/11,850	49/15,150	0.0008
		6.1 per 1000 days	3.2 per 1000 days	0.53 (0.37–0.77)
CONS	CONS	49	20	<0.001
Staph aureus	MSSA	10	14	0.99
	MRSA	2	0	-
VGS	VGS	2	4	0.91
Other Gram-positive bacteria		8	14	0.62
		71/11,850	52/15,150	0.003
		6 per 1000 days	3.4 per 1000 days	0.57 (0.40–0.82)
Fungi	Candida	6	6	0.89
	Other	1	0	-
		7/11,850	6/15,150	0.66
		0.6 per 1000 days	0.4 per 1000 days	0.67 (0.23–1.99)
		150/11,850	107/15,150	<0.0001
		12.7 per 1000 days	7.1 per 1000 days	0.56 (0.44–0.72)

## Data Availability

The data are available upon request from the corresponding author.

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
