# Peer review of "The Effectiveness of Taurolidine Antimicrobial Locks in Preventing Catheter-Related Bloodstream Infections (CRBSIs) in Children Receiving Parenteral Nutrition: A Case Series"

_antibiotics, 2024, doi:10.3390/antibiotics13090847_

Round 1

Reviewer 1 Report

Comments and Suggestions for Authors

Ling et al report the efficacy of taurolidine lock (TL) in preventing central-line associated bloodstream infections and related hospitalizations in children with parenteral nutrition  in the home setting.

The data are interesting and important for pediatricians caring for children with long term central lines. Taurolidine seems to be promising for the reduction of CLABSI also in children.

Nevertheless there are major issues that have to be solved before it is considered for publication:

One of the major problem is the incorrect definition of CLABSI. This incorrect definition leads to confusion. The authors should adhere to the standard definitions of CLABSI and CRBSI. CLABSI is a primary laboratory confirmed bloodstream infection in a patient with a central line at the time of (or within 48-hours prior to) the onset of symptoms and the infection is not related to an infection from another site. Instead the authors are referring to 2 blood cultures (that should become positive with a difference of at least 2 hours), that is the definition of catheter Related Blood Stream Infection (CRBSI).

One other major problem is the definition of the episodes and hsopitalisations

Every single episode should be counted as an episode. Incidence of infections should be reported as n° of episodes/ days of central line.

Why are two episodes occurring in the same month considered as the same episode? And also why were several hospitalisations counted as one month?

Furthermore, the design of the study is not clear :   It is a retrospective study  and two periods are considered; before 2020, when no TL locks were used, and after 2020 when they started using TL Locks. In the methods section it has to be better clarified that the authors use as control and treatment the same patient: a same patient is included in both periods: pre and post introduction of TL

Line 96: I would not say “during these follow up months” as it was not a prospective inclusion. I would say during the following months after inclusion.

It would be interesting to have the standardized training protocol for home care (even in supplemental material)

The authors should better explain how the lock was performed:  Was the lock performed every day? Was taurolidine withdrawn or flushed at the end of the lock?

Line 130 “Hospitalization episode was recorded when a culture was obtained during a hospital  stay of at least two days, with or without a CLABSI diagnosis.” Once again the definition of CLABSI is wrong: how can there be a positive culture but no diagnosis of CLABSI? Unless the culture was considered positive because of another site infection, but these should not be considered a CLABSI nor counted as an episode!

Once again why “Several hospitalizations within a month were counted as a single hospitalization episode”?

Line 168 “During this period, the range of CLABSI episodes rate was 0 and 28 episodes” I don’t understand what the authors mean.  Is this the min and max range?

Incidence of CLABSI should be reported as number of infections/1000 catheter days.

Author Response

Ling et al report the efficacy of taurolidine lock (TL) in preventing central-line associated bloodstream infections and related hospitalizations in children with parenteral nutrition  in the home setting. The data are interesting and important for pediatricians caring for children with long term central lines. Taurolidine seems to be promising for the reduction of CLABSI also in children. Nevertheless there are major issues that have to be solved before it is considered for publication: 

We thank the reviewer. We are committed to thoroughly revising our work to resolve these concerns and ensure it meets the high standards required for publication. We appreciate your guidance and will carefully consider each point in our revisions.

One of the major problem is the incorrect definition of CLABSI. This incorrect definition leads to confusion. The authors should adhere to the standard definitions of CLABSI and CRBSI. CLABSI is a primary laboratory confirmed bloodstream infection in a patient with a central line at the time of (or within 48-hours prior to) the onset of symptoms and the infection is not related to an infection from another site. Instead the authors are referring to 2 blood cultures (that should become positive with a difference of at least 2 hours), that is the definition of catheter Related Blood Stream Infection (CRBSI).

We appreciate this insightful comment. To improve the clarity of our article, we have revised the terminology throughout to strictly adhere to the standard definitions of CLABSI and CRBSI. Specifically, in the Methods section, we have updated the definition of CRBSI as outlined by the Infectious Diseases Society of America (IDSA). It now reads: “A diagnosis of a CRBSI episode was made whenever a patient presented with a positive blood culture. According to the Infectious Diseases Society of America (IDSA) definition, the growth of microorganisms had to be shown from at least two blood culture sets taken from the catheter and peripheral blood, as there is no infection of another site.[25] Although the gold standard for diagnosing CRBSI is a culture of the catheter tip, this was not included as a diagnostic criterion since we aimed to preserve the line whenever possible.[22]”

One other major problem is the definition of the episodes and hsopitalisations. Every single episode should be counted as an episode.

We thank the reviewer for their valuable comment. Given the complex nature of these patients, all of whom struggle with intestinal failure, hospitalizations are typically prolonged, and the healing process is challenging. To simplify the presentation and enhance reader understanding, we chose to present the number of episodes per month, as no significant differences were observed. Additionally, it can be unclear whether two separate episodes occurred, as cultures often remained positive.

Incidence of infections should be reported as n° of episodes/ days of central line. 

Done.

Why are two episodes occurring in the same month considered as the same episode? And also why were several hospitalisations counted as one month? 

We thank the reviewer for their valuable comment. Given the complex nature of these patients, all of whom struggle with intestinal failure, hospitalizations are typically prolonged, and the healing process is challenging. To simplify the presentation and enhance reader understanding, we chose to present the number of episodes per month, as no significant differences were observed. Additionally, it can be unclear whether two separate episodes occurred, as cultures often remained positive.

Furthermore, the design of the study is not clear :   It is a retrospective study  and two periods are considered; before 2020, when no TL locks were used, and after 2020 when they started using TL Locks. In the methods section it has to be better clarified that the authors use as control and treatment the same patient: a same patient is included in both periods: pre and post introduction of TL

Done. It is now reads: “For each patient, the follow-up period was divided into pre- and post-treatment with TL, with each patient serving as their own control, being included in both periods.”

Line 96: I would not say “during these follow up months” as it was not a prospective inclusion. I would say during the following months after inclusion. 

Done.

It would be interesting to have the standardized training protocol for home care (even in supplemental material).

In response to the reviewer's request, here is a brief overview of the standardized training protocol for home care as outlined in the manufacturer's TauroLockTM catalog:

  1. Follow the manufacturer’s instructions for the specific venous access device.
  2. Flush the device with 10 mL of saline.
  3. Withdraw TauroLockTM using an appropriate syringe.
  4. Instill TauroLockTM slowly (up to 1 mL per second; for infants and children under two years, up to 1 mL per 5 seconds) to fill the lumen completely. Adhere strictly to the manufacturer's specified fill volume.
  5. TauroLockTM remains in the device until the next treatment (up to 30 days).
  6. Before the next treatment, aspirate and discard TauroLockTM per institutional policy. If aspiration is not possible, flush slowly (up to 1 mL per 3 seconds). In rare cases where aspiration is not advised, slow flushing is acceptable, except in infants and children under two years.
  7. Flush the device with 10 mL of saline.

The "Taurolidine Lock Treatment Protocol" has been edited according to the reviewer's recommendations and now reads as follows: All patients were guided to follow the manufacturer instructions that accompany the particular venous vascular access product utilized. Specific catheter lock volumes are associated with each device. The treatment is provided by the caregiver daily after parenteral nutrition.  The treatment steps are briefly described as follows: 1. Flush the device with 10 mL of saline to ensure patency; 2. Withdraw TauroLock™ from the container using an appropriate syringe; 3. Instill TauroLock™ slowly (not more than 1 mL per second; for infants and children less than two years of age, not more than 1 mL per 5 seconds) into the access device in a quantity sufficient to fill the lumen completely.4. TauroLockTM remains in the device until the next treatment 5. Flush the device with 10 mL of saline.”

The authors should better explain how the lock was performed:  Was the lock performed every day? Was taurolidine withdrawn or flushed at the end of the lock?

Done. A sentence has been added to the Methods section: “The treatment is provided by the caregiver daily after parenteral nutrition.”

Line 130 “Hospitalization episode was recorded when a culture was obtained during a hospital  stay of at least two days, with or without a CLABSI diagnosis.” Once again the definition of CLABSI is wrong: how can there be a positive culture but no diagnosis of CLABSI? Unless the culture was considered positive because of another site infection, but these should not be considered a CLABSI nor counted as an episode! 

We thank the reviewer for their comment. To improve the clarity of our article, we have revised the terminology throughout to strictly adhere to the standard definitions of CLABSI and CRBSI. “A hospitalization episode was recorded when a culture was obtained during a hospital stay of at least two days, with or without a CRBSI diagnosis.” This means that a positive culture from the catheter (CRBSI) was not necessarily required. The aim was to capture the overall number of hospitalizations that raised clinical suspicion of catheter-related infection (possibly with another source), given the significant burden on both the patient and their family, as well as the healthcare system.

Once again why “Several hospitalizations within a month were counted as a single hospitalization episode”?

We thank the reviewer. We addressed this important comment previously.

Line 168 “During this period, the range of CLABSI episodes rate was 0 and 28 episodes” I don’t understand what the authors mean.  Is this the min and max range?

The sentence was rewritten for clarity and now reads: “During this period, the range of CRBSI episodes rate was 0 and 28 episodes, representing the minimum and maximum number of episodes observed (median 2.5 episodes, mean of 5.3 ± 7.7 days).”

Incidence of CLABSI should be reported as number of infections/1000 catheter days. 

Done.

Reviewer 2 Report

Comments and Suggestions for Authors

This is a very nice retrospective study which deserves publication.

I would only ask the authors to comment about their choice of using a taurolidine+citrate lock (TauroLock = 1.35% taurolidine + 4% citrate) rather than a plain taurolidine lock (NutriLock or Taurosept = 2% taurolidine), considering that (a) adding citrate to the lock is not strictly needed as anticoagulant, considering that long term venous access devices used for parenteral nutrition do not require anticoagulant, according to all current guidelines, (b) 4% citrate does not increase the antibacterial activity of the lock, since citrate has relevant antibacterial activity only >10%, and (c) citrate - according to the literature - may interact with the parenteral nutrients and cause lumen occlusion. I feel that the choice of a 2% taurolidine lock would have been more reasonable and more obvious. 

Author Response

This is a very nice retrospective study which deserves publication.

We thank the reveiwer.

I would only ask the authors to comment about their choice of using a taurolidine+citrate lock (TauroLock = 1.35% taurolidine + 4% citrate) rather than a plain taurolidine lock (NutriLock or Taurosept = 2% taurolidine), considering that (a) adding citrate to the lock is not strictly needed as anticoagulant, considering that long term venous access devices used for parenteral nutrition do not require anticoagulant, according to all current guidelines, (b) 4% citrate does not increase the antibacterial activity of the lock, since citrate has relevant antibacterial activity only >10%, and (c) citrate - according to the literature - may interact with the parenteral nutrients and cause lumen occlusion. I feel that the choice of a 2% taurolidine lock would have been more reasonable and more obvious. 

We appreciate the reviewer’s insightful comment and agree that a 2% taurolidine lock could be a reasonable alternative. Our use of a taurolidine+citrate lock (TauroLock = 1.35% taurolidine + 4% citrate) instead of a plain taurolidine lock (NutriLock or Taurosept = 2% taurolidine) is based on its availability from our supplier and the desire to enhance safety against thrombus formation. Although current guidelines do not mandate anticoagulants for long-term venous access devices for parenteral nutrition, the citrate was included to complement the antibacterial effects of taurolidine and align with institutional protocols. We are monitoring its impact to ensure it meets safety and efficacy standards.

A sentence was added to the Methods section and is now reads:We used TauroLock™ (1.35% taurolidine + 4% citrate) due to its availability and to enhance thrombus safety. While guidelines do not require anticoagulants for parenteral nutrition devices, citrate complements the antibacterial effects of taurolidine and aligns with institutional protocols.”

Reviewer 3 Report

Comments and Suggestions for Authors

Methods

'a pre- and post-treatment' (line 95) is not 'A retrospective case-series study' (line 14).

Results

The author reported 14 patients (line 159). How many patients are in the pre-treatment period and the post-treatment period?

Comments on the Quality of English Language

moderate

Author Response

Methods

'a pre- and post-treatment' (line 95) is not 'A retrospective case-series study' (line 14).

 We thank the reviewer. For clarity, we have added the following sentence to the Methods section: “For each patient, the follow-up period was divided into pre- and post-treatment with TL, with each patient serving as their own control, being included in both periods.”

Results

The author reported 14 patients (line 159). How many patients are in the pre-treatment period and the post-treatment period?

Each of the 14 patients served as their own control, comparing the pre-treatment period to the post-implantation period of TauroLock antimicrobials. For clarity, we have added the following sentence to the Methods section: “For each patient, the follow-up period was divided into pre- and post-treatment with TL, with each patient serving as their own control, being included in both periods.”

Comments on the Quality of English Language- moderate

The language has been revised and refined to enhance the readability of the article.

Round 2

Reviewer 3 Report

Comments and Suggestions for Authors

Reviewer comment

Manuscript ID: antibiotics-3162308-peer-review-v2

My apologies for not giving a good explanation. I don't understand the study protocol. Is what I think about this correct? There are 14 patients from January 2017. They were treated with PN without TL. The author records the per-TL outcome (bacterial culture, hospitalization episode). The same (14) patients were admitted after 2020 and treated with TL. The author records the outcome as post-TL.

Page 2, lines 88-90: Taurolidine lock was incorporated into our center since June 2020. Therefore, we set the follow-up period from January 2017, three years prior to the incorporation of the treatment, to June 2024.

Page 2, lines 92-93: Patients included in the study were children with intestinal failure, treated with PN via a central line after 2017, and administered TL.

Page 3, lines 95-96: For each patient, the follow-up period was divided into pre- and post-treatment with TL, with each patient serving as their own control, being included in both periods.

How long ‘TauroLockTM remains in the device until the next treatment’? Page 3, lines 114-115: TauroLockTM remains in the device until the next treatment.

What is the study's implication? The TL has been a hospital policy since 2020. Page 2, lines 88-90: Taurolidine lock was incorporated into our center since June 2020.

Is this TL policy not for other types of central venous access devices, including short-term and intermediate-term devices (Page 8, lines 65-66)?

Comments on the Quality of English Language

Moderate

Author Response

My apologies for not giving a good explanation. I don't understand the study protocol. Is what I think about this correct? There are 14 patients from January 2017. They were treated with PN without TL. The author records the per-TL outcome (bacterial culture, hospitalization episode). The same (14) patients were admitted after 2020 and treated with TL. The author records the outcome as post-TL. Page 2, lines 88-90: Taurolidine lock was incorporated into our center since June 2020. Therefore, we set the follow-up period from January 2017, three years prior to the incorporation of the treatment, to June 2024.

Page 2, lines 92-93: Patients included in the study were children with intestinal failure, treated with PN via a central line after 2017, and administered TL.

Page 3, lines 95-96: For each patient, the follow-up period was divided into pre- and post-treatment with TL, with each patient serving as their own control, being included in both periods.

We thank the reveiwer. The study protocol has been revised for clarity and it is now reads: “This retrospective case series study assessed all children with intestinal failure who were administered PN and treated with TL at the SUMC pediatric day-care unit, between 2017 and 2024 (N=14). Patient characteristics, culture results, and clinical parameters were obtained from the medical files and entered into a computerized database. For each patient, the follow-up period was divided into pre- and post-treatment with TL, with each patient serving as their own control, being included in both periods. Taurolidine lock was incorporated into our center since June 2020. Therefore, we set the follow-up period from January 2017, three years prior to the incorporation of the treatment (pre-treatment period), to June 2024 (post-treatment period). During these follow up months, episodes of CRBSI were counted and analyzed, including culture results and hospitalizations.”

How long ‘TauroLockTM remains in the device until the next treatment’? Page 3, lines 114-115: TauroLockTM remains in the device until the next treatment.

This sentence has been modified and now reads: “ TauroLockTM remains in the device on daily basis until the subsequent treatment”

What is the study's implication? The TL has been a hospital policy since 2020. Page 2, lines 88-90: Taurolidine lock was incorporated into our center since June 2020.

This study highlights the impact of taurolidine (TL) antimicrobial locks incorperation to our hospital in reducing catheter-related bloodstream infections (CRBSI) among children with intestinal faliure receiving parenteral nutrition (PN). Over a period from 2017 to 2024, the introduction of TL resulted in a 45% reduction in bacterial CRBSI rates and a 41% decrease in related hospitalizations.  As for Antimicrobial effect, we found that TL was effective against both Gram-negative and Gram-positive bacteria, though its impact on fungal infections was less pronounced. The findings underscore the potential of TL as an effective strategy for preventing CRBSI in pediatric patients with intestinal failure, advocating for its broader application in similar patient populations.

Is this TL policy not for other types of central venous access devices, including short-term and intermediate-term devices (Page 8, lines 65-66)?

In the Soroka University Medical Center (SUMC), we use taurolidine locks (TL) only for the indications presented in our study. To the best of our knowledge, data on the efficacy of TL for short-term and intermediate-term devices is sparse and has mostly been presented through meta-analyses of different types of catheters.[1,2] Although some research indicates a reduced CRBSI rate with TL antimicrobials, a more focused study on short-term and intermediate-range devices is needed to achieve higher confidence in these findings.

References

  1. Liu Y, Zhang AQ, Cao L, Xia HT, Ma JJ. Taurolidine Lock Solutions for the Prevention of Catheter-Related Bloodstream Infections: A Systematic Review and Meta-Analysis of Randomized Controlled Trials. PLoS One [Internet]. 2013 Nov 21 [cited 2024 Sep 2];8(11). Available from: /pmc/articles/PMC3836857/
  2. van den Bosch CH, Jeremiasse B, van der Bruggen JT, Frakking FNJ, Loeffen YGT, van de Ven CP, van der Steeg AFW, Fiocco MF, van de Wetering MD, Wijnen MHWA. The efficacy of taurolidine containing lock solutions for the prevention of central-venous-catheter-related bloodstream infections: a systematic review and meta-analysis. Journal of Hospital Infection. 2022 May 1;123:143–55.